# Indole- and Pyrazole-Glycyrrhetinic Acid Derivatives as PTP1B Inhibitors: Synthesis, In Vitro and In Silico Studies

**DOI:** 10.3390/molecules26144375

**Published:** 2021-07-20

**Authors:** Ledy De-la-Cruz-Martínez, Constanza Duran-Becerra, Martin González-Andrade, José C. Páez-Franco, Juan Manuel Germán-Acacio, Julio Espinosa-Chávez, J. Martin Torres-Valencia, Jaime Pérez-Villanueva, Juan Francisco Palacios-Espinosa, Olivia Soria-Arteche, Francisco Cortés-Benítez

**Affiliations:** 1Departamento de Sistemas Biológicos, División de Ciencias Biológicas y de la Salud, Universidad Autónoma Metropolitana–Xochimilco (UAM–X), Ciudad de México 04960, Mexico; 2193800101@alumnos.xoc.uam.mx (L.D.-l.-C.-M.); 2163025430@alumnos.xoc.uam.mx (C.D.-B.); jpvillanueva@correo.xoc.uam.mx (J.P.-V.); jpalacios@correo.xoc.uam.mx (J.F.P.-E.); soriao@correo.xoc.uam.mx (O.S.-A.); 2Maestría en Ciencias Farmacéuticas, División de Ciencias Biológicas y de la Salud, Universidad Autónoma Metropolitana–Xochimilco (UAM–X), Ciudad de México 04960, Mexico; 3Departamento de Bioquímica, Facultad de Medicina, Universidad Nacional Autónoma de México, Ciudad de México 04510, Mexico; martin@bq.unam.mx; 4Red de Apoyo a la Investigación, Universidad Nacional Autónoma de México e Instituto Nacional de Ciencias Médicas y Nutrición Salvador Zubirán, Ciudad de México 14000, Mexico; paez@cic.unam.mx (J.C.P.-F.); jmga@cic.unam.mx (J.M.G.-A.); 5Instituto de Investigaciones Químico Biológicas, Universidad Michoacana de San Nicolás de Hidalgo, Morelia 58030, Mexico; julioespinosa30@hotmail.com; 6Área Académica de Química, Universidad Autónoma del Estado de Hidalgo, Pachuca 42184, Mexico; jmartin@uaeh.edu.mx

**Keywords:** protein tyrosine phosphatase 1B, glycyrrhetinic acid, *p*-nitrophenylphosphate assay, molecular docking, molecular dynamics

## Abstract

Regulating insulin and leptin levels using a protein tyrosine phosphatase 1B (PTP1B) inhibitor is an attractive strategy to treat diabetes and obesity. Glycyrrhetinic acid (GA), a triterpenoid, may weakly inhibit this enzyme. Nonetheless, semisynthetic derivatives of GA have not been developed as PTP1B inhibitors to date. Herein we describe the synthesis and evaluation of two series of indole- and *N*-phenylpyrazole-GA derivatives (**4a**–**f** and **5a**–**f**). We measured their inhibitory activity and enzyme kinetics against PTP1B using *p*-nitrophenylphosphate (pNPP) assay. GA derivatives bearing substituted indoles or *N*-phenylpyrazoles fused to their A-ring showed a 50% inhibitory concentration for PTP1B in a range from 2.5 to 10.1 µM. The trifluoromethyl derivative of indole-GA (**4f**) exhibited non-competitive inhibition of PTP1B as well as higher potency (IC_50_ = 2.5 µM) than that of positive controls ursolic acid (IC_50_ = 5.6 µM), claramine (IC_50_ = 13.7 µM) and suramin (IC_50_ = 4.1 µM). Finally, docking and molecular dynamics simulations provided the theoretical basis for the favorable activity of the designed compounds.

## 1. Introduction

For many years, protein tyrosine phosphatase 1B (PTP1B) has been known for its role in the negative regulation of the insulin and leptin signaling pathway [1,2,3,4]. Therefore, blocking this enzyme is an attractive strategy to treat both diabetes and obesity, which represents an advantage over existing therapies to date. Furthermore, PTP1B has gained much attention in recent years since it is involved in other diseases such as Alzheimer’s [5], Parkinson’s disease [6,7], cardiovascular disorders [8,9,10,11], and malignant tumors [12,13,14,15,16]. This makes the development of PTP1B inhibitors even more interesting.

Extensive research has demonstrated that natural oleanane-type pentacyclic triterpenes display potent inhibitory activity and selectivity against PTP1B [17,18,19,20,21]. Maslinic acid [22,23], oleanolic acid [24,25,26] and celastrol [27,28,29] are the most representative examples of this type of inhibitors. Less known is glycyrrhetinic acid (GA), which is an oleanane-type triterpene found abundantly (up to 24%) as glycyrrhizin in the root of licorice (*Glycyrrhiza glabra* L.) [30,31,32]. GA has shown several beneficial pharmacological activities including anti-viral, anti-cancer, anti-inflammatory, anti-microbial, anti-fungal and hepatoprotective properties [33]. Furthermore, oral administration of 18β-GA (100 mg/kg) enhances insulin levels and reduces blood glucose levels comparable to that of glibenclamide in streptozotocin-induced diabetic rats [34,35]. Recently it has been demonstrated that GA improves glucose uptake and reverses insulin resistance by targeting RAS proteins and activating the PI3K/Akt pathway, respectively [36]. In a patent application, GA was also shown to reduce the lipid synthesis or formation of very-low-density lipoproteins (VLDL’s) [37]. Moreover, both GA epimers (18α and 18β) have been identified as weak competitive PTP1B inhibitors [38,39]. Interestingly, GA can cross the blood-brain barrier [40,41] which is an essential property in the context of PTP1B inhibitors. Reducing the hypothalamic PTP1B activity of obese mice has been shown to greater enhance leptin and insulin sensitivity than that of other tissues, which also effectively decreases adiposity and improves glucose metabolism [3,42]. Hence, we believe that GA is an attractive scaffold for further chemical modifications to improve its PTP1B-inhibitory activity. 

Accordingly, it should be noted that the semisynthesis of ring-substituted triterpenes having nitrogenated heterocycles fused to the A-ring has been reported as a strategy for improving PTP1B inhibitory activity [19,20]. For instance, combining the indole or *N*-phenylpyrazole scaffold to maslinic acid (MA, an olenane-type triterpene) has dramatically improved the potency of this triterpenoid [43]. Moreover, the presence of indole or *N*-phenylpyrazole moieties at the A-ring of MA produced more potent PTP1B inhibitors than other heterocycles such as thiazole, pyrimidine, quinazoline and pyrazine. Therefore, this strategy could be applied to GA, but it remains to be validated, since no attempt of semisynthetic heterocyclic ring-substituted GA derivatives as PTP1B inhibitors has been reported to date. Thus, we focused on increasing the inhibitory activity of GA, which is relatively easy and cheap to obtain in large quantities from plant extracts. To do so, we synthesized two series of GA derivatives (Figure 1): series one regroups six A-ring-fused indoles, whereas series two regroups six A-ring-fused *N*-phenylpyrazoles. It is well-known that the catalytic site of PTP1B is a positively-charged pocket formed by side chains of Tyr^46^, Arg^45^, Asp^48^, Lys^120^, Phe^182^, Ile^219^, Asp^181^ and Gln^262^ [44], whereas the allosteric binding site has a hydrophobic pocket formed by side chains of Leu^192^, Asn^193^, Phe^196^ Glu^276^, Lys^279^, and Phe^280^ [45]. From these observations, we hypothesized that introducing an atom or group of atoms bearing a partially negative charge (e.g., -OCH_3_, -F, -C and -CF_3_), as well as hydrophobic groups (e.g., -CH_3_ and -CF_3_) on the phenyl moiety of indole (series one) and *N*-phenylpyrazole (series two), can improve dipole-dipole and hydrophobic interactions at both catalytic and allosteric binding sites, respectively. Moreover, the size of these groups could promote an ideal orientation for the designed compounds to perform π-stacking and hydrophobic interactions between the phenyl moiety and the side chains of aromatic amino acids at the allosteric binding site.

The newly synthesized compounds were tested at several concentrations against PTP1B. Their inhibitory activity was compared to the reference compounds carbenoxolone (CA), ursolic acid (UA), claramine (CL) and suramin (SU) (Figure 1). UA is a natural ursane-type pentacyclic triterpenoid that has been identified as a potent PTP1B inhibitor. Moreover, this compound enhances insulin receptor signaling and stimulates glucose uptake in vitro [46]. The second, CL, is a potent allosteric PTP1B inhibitor that has been shown to restore glycemic control and cause weight loss without increasing energy expenditure in mice. Due to its similar structure to trodusquemine, it has been proposed that CL targets the disordered C terminus of PTP1B [47,48]. Finally, SU is an anti-trypanosomal drug used to treat African sleeping sickness and river blindness. However, it has shown potent competitive PTP1B inhibitory activity [49]. On the other hand, despite the PTP1B inhibitory activity of CA (the hemisuccinate derivative of GA), it has not been reported on to date; thus, we decided to assess this compound because of its similar structure to GA. CA has been widely used to treat gastric ulcers and other types of inflammation by targeting 11β-hydroxysteroid dehydrogenase (11β-HSD). It has also shown amelioration of metabolic syndrome in obese rats [50].

## 2. Results and Discussion

### 2.1. Chemistry

The synthetic route for the synthesis of indole- and pyrazole-GA derivatives is shown in Scheme 1. Briefly, GA was oxidized to ketone (**2**) using Jones reagent. Indole-GA derivatives (**4a**–**f**) were prepared using Fischer indolization by reacting compound **2** with phenylhydrazine or *p*-substituted phenylhydrazines in refluxing acetic acid. It is worth mentioning that these final compounds were obtained in good overall yields. On the other hand, the intermediate **2** was reacted with ethyl formate in the presence of NaH to afford compound **3**. Afterwards, this 1,3-dicarbonylic intermediate was reacted with the corresponding phenylhydrazine or *p*-substituted phenylhydrazines to afford the *N*-phenylpyrazole-GA derivatives (**5a**–**f**) in moderate yields. 

### 2.2. NMR Characterization of Compounds ***4f*** and ***5f***

The sequence of reactions used herein made it possible to obtain series **4a**–**f** and **5a**–**g**. However, it was necessary to confirm that the indole and pyrazole heterocycles were fused to the A-ring of GA. Therefore, to obtain a full assignation for H and C atoms, we carried out an analysis for the most representative compounds **4f** and **5f** (Figure 2) using ^1^H NMR and ^13^C NMR spectra along with COSY, HSQC, HMBC and NOESY experiments (Appendix A). 

For both analogs, the chemical shift data of the GA skeleton is in accordance with previous studies [51,52]. However, in the ^1^H NMR spectra, there are some representative signals in the aromatic region from 7.2 to 7.9 ppm. The two doublets (*J* = 7.8 and 7.9 Hz) characteristic of a *para* substituted pattern, were assigned to the phenyl ring of compound **5f**, whereas a singlet at 7.36 ppm corresponded to the pyrazole ring. On the other hand, two singlets with different intensities at 7.79 ppm (H–4′) and 7.34 ppm (H–6′ and H–7′) for **4f** were found; these signals along with a broad singlet at 7.91 ppm (NH) correspond to the indole ring. 

When the indolization reaction was carried out, the signal at about 217 ppm (C3 carbonyl for compound **2**) is shifted upfield for compound **4f** at 142.1 ppm in the ^13^C NMR spectra. Similarly, when *N*-phenylpyrazole was prepared, both signals at 200 and 210 ppm (C3 carbonyl and formyl at C2, for intermediate **3**) are shifted upfield for compound **5f** at 145.79 and 114.4 ppm, respectively. Using an HMBC experiment, we first identified the methyl groups that produced correlation with the C3 signal in both **4f** and **5f**. The correlations observed with 23–CH_3_ and 24–CH_3_ in combination with ^13^C NMR data found in literature allowed the identification of the following carbons: 1–CH_2_, 2–C, 4–C, 5–CH, 10–C. Using an HSQC experiment, the 5–C signals observed at 53.1 ppm (for **4f**) or 53.58 ppm (for **5f**) were linked to the 5α–H at 1.44 ppm (for **4f**) and 1.20 ppm (for **5f**), respectively. In the HMBC experiment, the 5α–H signal allowed the identification of 25β–CH_3_ which correlates with a signal at about 37 ppm (for both **4f** and **5f**) as well as 108 ppm (for **4f**) and 114 ppm (for **5f**). These signals correspond to 1–C and 2–C carbons. The 1–C signal was linked to both 1–CH_2_ hydrogens at 2.3 and 4.0 ppm for **4f** as well as 2.2 and 3.6 ppm for **5f**. The signal at 2.3 ppm (for **4f**) and 2.2 ppm (for **5f**) produces a NOE correlation with 25β–CH_3_; hence, it corresponds to 1β–H, whereas the signal at 4.0 ppm (for **4f**) and 3.6 ppm (for **5f**) corresponds to 1α–H. It is worth mentioning that both 1–CH_2_ signals are shifted downfield compared to compound **2**, which is possible with the presence of heterocycles fused to the C2 and C3 positions of the triterpenoid skeleton. 

### 2.3. PTP1B Inhibitory Activity of Indole- and Pyrazole-GA Derivatives

Compounds **4a**–**f** and **5a**–**f** were tested against PTP1B enzyme to determine their ability to inhibit the formation of *p*-nitrophenol from *p*-nitrophenylphosphate. Different concentrations of **4a**–**f** and **5a**–**f**, as well as reference inhibitors **GA**, **CA**, **UA**, **SU** and **CL**, provided the IC_50_ values shown in Table 1. The first general observation from these data is that the fusion of both indole and *N*-phenylpyrazole heterocycles to the GA skeleton yielded potent PTP1B inhibitors (IC_50_ values ranging from 10.1 to 2.5 µM) with 6 to 25-fold better potency than GA (IC_50_ = 62.0 µM). The newly-designed compounds exhibited better inhibitory effect against PTP1B than that shown by positive controls **UA** (IC_50_ = 5.6 µM), **CA** (IC_50_ = 69.7 µM) and **CL** (IC_50_ = 13.7 µM). Precisely, the order of potency for indole–GA inhibitors was **4f** > **4b** > **4d** > **4a** > **4c** > **4e** whereas for *N*-phenylpyrazole–GA inhibitors, the order was **5f** > **5b** = **5c** = **5a** > **5e** > **5d**. Thus, the substituent attached to the phenyl moiety modulates the PTP1B inhibitory activity. In this way, we noticed that introducing a chlorine atom or a methyl group at the C5′ position of the indole ring has a negative impact on activity compared to **4a** (IC_50_ = 6.6 µM). Conversely, introducing a fluorine, methoxy or trifluoromethyl group at the same position greatly increases the potency. Indeed, **4f** (IC_50_ = 2.5 µM) is the most potent PTP1B inhibitor among the two series. It is also worth noting that this inhibitor is the only one that showed better potency than that of **SU** (IC_50_ = 4.1 µM). On the other hand, for *N*-phenylpyrazole–GA inhibitors, the presence of fluorine (**5d**: IC_50_ = 9.6 µM) and chlorine (**5e**: IC_50_ = 7.5 µM) at the phenyl moiety is not well tolerated when compared to the unsubstituted derivative **5a** (IC_50_ = 5.1 µM). Contrarily, methoxy (**5b**: IC_50_ = 4.8 µM), methyl (**5c**: IC_50_ = 4.8 µM) and trifluoromethyl (**5f**: IC_50_ = 4.4 µM) groups improve the inhibitory activity against PTP1B. 

According to these findings, we believe that indole–GA derivatives bearing electronegative as well as HBA groups improve their inhibitory activity against PTP1B. In contrast, bulky hydrophobic groups attached to *N*-phenylpyrazole–GA derivatives had a positive impact on their potency. Furthermore, it is important to mention that the incorporation of trifluoromethyl groups significantly enhanced the inhibitory effect of both series.

### 2.4. Enzymatic Kinetic Studies

Some compounds were chosen to perform enzyme kinetic assays to investigate their type of inhibition. Kinetic analyses were performed at different concentrations of substrate and inhibitor. Table 2 and Figure 3 show the kinetic parameters and the graphic representation (Lineweaver-Burk plot). Compounds **4f**, **5f**, **UA**, and **CL** show non-competitive inhibition (K_i_ values: 3.9, 4.6, 8.9, and 15.4 µM, respectively), and only **SU** shows competitive inhibition (K_i_ value = 7.1 µM). Knowing the type of inhibition of the enzyme–inhibitor complex is important to identify potential bioactive molecules to develop therapeutic agents and direct molecular modeling studies.

### 2.5. Molecular Docking Studies

To investigate the potential binding modes of **4a**–**f** and **5a**–**f**, we carried out a consensus docking simulation using Autodock (AD), Autodock Vina and GOLD. This study was performed employing the three aforementioned docking softwares to combine the information of multiple scoring functions to identify the correct or most accurate solution for each ligand–protein interaction. The designed compounds and controls were docked at the catalytic site (PDB ID: 1C83 [44]) and the experimentally validated allosteric site (PDB ID: 1T49 [45]) using the X–ray structures of PTP1B. Since docking validation is an important aspect for measuring the quality of the docking protocol, we re-docked the co-crystal structures found in both 1C83 and 1T49 complexes. To do so, 6-(oxalyl-amino)-1*H*-indole-5-carboxylic acid (AOI) was docked into the active site (PDB ID: 1C83) while 3-(3,5-dibromo-4-hydroxybenzoyl)-2-ethyl-benzofuran-6-sulfonic acid (4-sulfamoyl-phenyl)-amide (892) was docked at the allosteric binding site (PDB ID: 1T49). The resulting docking simulations for both ligands (Table 3) provided lower binding free energies in AD and Vina as well as higher CHEMPLP fitness score (CS) and GoldScore (GS) values in GOLD, indicating a favorable ligand binding affinity to their respective binding sites. 

After superimposing the binding pose of the re-docked structure onto the binding pose of the co-crystal structure, a root mean square deviation (RMSD) value of less than 1.5 Å was obtained. We therefore believe that the docking protocol presented herein is a good reproduction of the correct binding for the co-crystal ligands. Hence, it may be efficient to predict the most favorable binding pose for the designed compounds and controls.

As shown in Table 4, **SU** fitted better at the catalytic site, providing lower binding free energies (−15.6 and −9.2 kcal/mol for AD and Vina respectively) than those of the allosteric binding site (−11.2 and −7.2 kcal/mol for AD and Vina, respectively). Indeed, these data are in accordance with our enzymatic kinetic studies where **SU** is a competitive inhibitor. Similarly, GA and CA showed more favorable binding energies and scores at the catalytic site. Conversely, UA displayed similar binding energies as well as scores at both catalytic and allosteric binding sites of PTP1B. Therefore, we cannot discern for which does it have more affinity. As expected, the synthesized derivatives (**4a**–**f** and **5a**–**f**) showed better binding free energies and scores than those of reference compounds **GA**, **CA** and **UA** at both the catalytic and allosteric binding sites of PTP1B. Nevertheless, most of these compounds yielded lower binding free energies as well as higher scores for the allosteric binding site, which indicates that they preferably interact with this one; this in turn may explain why compounds **4f** and **5f** are non-competitive inhibitors. Precisely, the triterpenoid **4f** gave better binding free energy values and scores [−8.8 kcal/mol (AD), −9.9 kcal/mol (Vina), 59.5 (CS) and 49.6 (GS)] compared to **5f** [−8.3 kcal/mol (AD), −8.2 kcal/mol (Vina), 43.7 (CS) and 28.5 (GS)]. Interestingly, both analogs fit with the same orientation into the allosteric binding site (Figure 4). For instance, the pyrazole and indole moiety of **4f** and **5f**, respectively, perform key π-stacking interactions as well as hydrophobic contacts with the side chain of Phe^196^ and Leu^192^. Additionally, the triterpenoid skeleton of both derivatives also interacts by means of hydrophobic contacts with Phe^196^ residue. However, compound **4f** produces an H–bond between its NH group and the carbonyl group from the side chain of Asn^193^, while compound **5f** shows an extra π-stacking interaction between its phenyl moiety and the side chain of Phe^280^ which is absent in compound **4f**. Surprisingly, both compounds performed halogen interactions between their -CF_3_ group and Glu^276^ or Ala^189^. Of note, it has been demonstrated that the -CF_3_ group can enhance ligand binding affinity through close contacts with backbone carbonyl groups in a protein (orthogonal multipolar C–F···C=O interactions) [53]. Thus, this may explain why the -CF_3_ group significantly enhanced the PTP1B inhibitory activity in both GA series. From docking results, we consider that moieties able to perform π-stacking interactions (such as pyrazole and indole) play a critical role in improving the binding affinity into the allosteric binding site of PTP1B. 

### 2.6. Molecular Dynamics Simulations

Molecular dynamics simulations were performed using the *YASARA structure* software [54] and the AMBER14 [55] forcefield. Three different studies were carried out for the positive control **UA** and derivatives **4f** and **5f** resulting from the docking simulations at the allosteric binding site of PTP1B. In the first one, we evaluated RMSD fluctuations of the Cα carbons to know the stability of each protein–ligand complex. In the second, we calculated the binding affinity of each ligand at the allosteric site of PTP1B during a period of 50 nanoseconds of MD simulation. Finally, to find out the efficacy of the compounds **UA**, **4f** and **5f** to dissociate from the allosteric site of PTP1B, we carried out a steered molecular dynamics (SMD) simulation where a steering potential was applied to pull the ligand outside of the protein. 

#### 2.6.1. Root Mean Square Deviation (RMSD) Analysis

In this study, we performed an RMSD analysis on the PTP1B protein system and the PTP1B protein–ligand complexes of **UA**, **4f** and **5f** during 50 nanoseconds of time simulation. The results of this analysis can be employed as a measure of the overall fluctuation of the protein’s Cα carbons. Thus, the lower the value of RMSD, the less fluctuation of the Cα carbons there are, and therefore more stability for the protein’s main chain atoms. It is important to mention that PTP1B’s crystal structure used in this study (PDB ID: 1T49) exists as a catalytically inactive conformation. Thus, lower RMSD values indicate that the ligand stabilizes this form. As shown in Figure 5, the PTP1B system tends to stabilize after 3 nanoseconds with an average RMSD value = 1.02 Å (from 3 to 50 nanoseconds). However, slightly lower mean RMSD values were found in the PTP1B–UA and PTP1B–5f complexes (1.01 and 1.00 Å, respectively), which indicates that both compounds can stabilize the PTP1B protein system. At the same time, it can be clearly seen that the overall fluctuation of the PTP1B–4f complex is the lowest among the three (mean RMSD value = 0.91 Å), suggesting that the binding of **4f** at the allosteric site has the best impact for stabilizing the overall conformation of PTP1B in its inactive state.

#### 2.6.2. Binding Energy and Steered Molecular Dynamics (SMD) Studies

To quantitatively know the affinity of the three compounds to the allosteric binding site of PTP1B, we performed a binding energy analysis as a function of simulation time for a period of 50 nanoseconds including the binding energy average (Figure 6A). YASARA retrieves the binding energy by calculating the energy at an infinite distance between the selected object and the rest of the simulation system (i.e., the unbound state) and subtracting the energy of the simulation system (i.e., the bound state) every 100 picoseconds. Thus, more positive energy values indicate better binding [56] in the context of AMBER14 forcefield. Among the three simulated ligands, **UA** displayed an average binding energy value = −5.87 kcal/mol, whereas compounds **4f** and **5f** presented more positive binding energy profiles (average = −3.82 and −3.85 kcal/mol, respectively) suggesting better binding at this cavity. However, we noticed that **5f** showed less homogenous fluctuations in its binding energy profile compared to **UA** and **4f**. For instance, this compound exhibited strong negative as well as positive fluctuations at 16.4 nanoseconds (−21.13 kcal/mol) and 22.6 nanoseconds (29.5 kcal/mol) of MD simulation, respectively, which indicates less binding stability than **4f** at the allosteric binding site of PTP1B. 

To further provide information about the affinity of the ligands at the allosteric binding site, we carried out a SMD simulation. Of note, SMD is a special MD simulation where time-dependent external forces are applied to a ligand to facilitate its unbinding from a protein in order to study how both the ligand and the protein respond. The PTP1B complexes submitted to 50 nanoseconds MD simulations were used to pull off compounds **UA**, **4f** and **5f** to a distance of 15 Å from the allosteric binding site. The pulling was performed with a constant acceleration of 2000 pm/picoseconds^2^. As shown in Figure 6B, it can be clearly seen that compound **5f** dissociates earlier (23.6 ps) from the allosteric binding site than **4f** (44.4 ps). During the unbinding process of **UA** from PTP1B, we noticed that the carboxylate group of this triterpenoid formed a salt-bridge interaction with the ammonium group of Lys^197^. This strong interaction led to an increased residence time at this site. Thus, **UA** was found to be the last to leave the allosteric binding site (60.8 ps) among the three simulated ligands followed by compounds **4f** and **5f**. 

Overall, the results from MD and SMD simulations indicate that **4f**, **5f** and **UA** can bind to the allosteric binding site. However, **4f** is a more potent PTP1B inhibitor than **5f** and **UA** due to the fact that it better stabilizes the overall inactive conformation of PTP1B and also showed the most positive average binding energy value. Due to the slow dissociation of **UA** from the allosteric binding site, we believe that this compound can occupy this cavity for a longer time, thus stabilizing for a longer period PTP1B in its catalytically inactive conformation. Hence, this may explain why **UA** exhibited a potent PTP1B inhibitory effect even though it showed a less favorable binding energy profile than **4f** and **5f**.

## 3. Materials and Methods

### 3.1. General

All chemicals and starting materials were obtained from Sigma–Aldrich. (Toluca MEX, Mexico and St. Louis, MO, USA). Reactions were monitored by TLC on 0.2 mm percolated silica gel 60 F254 plates (Sigma–Aldrich) and visualized by irradiation with a UV lamp. Melting points were determined in a Fisher–Johns melting point apparatus and are uncorrected. Both ^1^H NMR and ^13^C NMR spectra were measured with Agilent DD2 (Agilent, Santa Clara, CA, USA) and Bruker Ascend (Bruker, Billerica, MA, USA) spectrometers, operating at 600 MHz and 400 MHz for ^1^H, respectively and 151 MHz for ^13^C. Chemical shifts are given in parts per million, relative to tetramethylsilane (Me_4_Si, δ = 0); *J* values are given in Hz. Splitting patterns are expressed as follows: s, singlet; d, doublet; q, quartet; dd, doublet of doublet; t, triplet; m, multiplet; bs, broad singlet. The numbering reported in Figure 2 was used for the assignment of ^1^H and ^13^C NMR signals. The UPLC–ESI–MS analyses for final compounds were performed on a Waters Acquity UPLC–HSS Class system (Waters, Wayland, MA, USA) equipped with a quaternary pump, sample manager, column oven, and photodiode array detector (PDA) interfaced with a SQD2 single-quadrupole mass spectrometer detector with an electrospray ion source. Empower software version 3 was used to control the UPLC–ESI–MS system and for data acquisition and processing. The analytical method was developed using an Acquity UPLC HSS T3 (3.0 × 100 mm, 8 μm). The mobile phase consisted of 0.1% aqueous formic acid (solvent A) and acetonitrile (solvent B), with an isocratic elution of 20% A and 80% of B. The flow rate was set to 0.7 mL/min, injection volume was 10 μL, and column temperature was maintained at 22 °C using a column oven. All samples were analyzed at 0.01 mg/mL. The detection was carried out through the run at 190–790 nm, while MS parameters were as follows: the cone and capillary voltages were set at 15.0 V and 0.8 kV, respectively; the source temperature was 600 °C. ESI–MS spectra were obtained using nitrogen as the collision gas, within a mass range of *m*/*z* 50–600 Da (scan duration of 0.5 s). Each sample was analyzed in positive (ESI+) mode. Compounds were named using the automatic name generator tool implemented in ChemDraw Professional 16.0.1.4 software (PerkinElmer, Waltham, MA, USA), according to IUPAC rules.

### 3.2. Synthesis

For the synthesis of GA compounds **4a**–**f** and **5a**–**f** (Scheme 1), the commercially available 18β–glycyrrhetinic acid was used. Compounds **2** and **3** were prepared by the procedure described in Refs. [57,58] with slight modifications. The synthesis of **4a**–**f** and **5a**–**f** is briefly described below.

#### 3.2.1. Synthesis of 3,11-Dioxo-olean-12-en-30-oic Acid (**2**)

A solution of 18β–glycyrrhetinic acid (1.5 g, 3.2 mmol) in acetone (100 mL) was cooled to 0 °C; CrO_3_ (688 mg, 6.9 mmol) in sulfuric acid (98%, 4.6 mL) and water (15.7 mL) was then added slowly. The mixture was stirred at room temperature overnight. The acetone was evaporated, and the residue was extracted with CH_2_Cl_2_ (3 × 50 mL) and washed with water and brine (1 × 50 mL). The combined organic layers were dried with sodium sulfate, filtered, and concentrated to dryness. Finally, the solid was recrystallized from MeOH, CH_2_Cl_2_, and drops of water to give **2** (1.37 g, 91%) as colorless crystals. M.p. ˃ 300 °C (308 °C [57]); ^1^H NMR (600 MHz, CDCl_3_): δ_H_ = 5.74 (s, 1H), 2.97 (m, 1H), 2.63 (m, 1H), 2.36 (ddd, *J* = 15.84, 6.4, 4.0 Hz, 1H), 2.20 (dd, *J* = 13.4 and 3.3 Hz, 1H), 2.01 (m, 2H), 1.93 (ddd, *J* = 13.7, 4.2, 2.7 Hz, 1H), 1.85 (d, *J* = 4.4 Hz, 1H), 1.72–1.32 (m, 13H), 1.37 (s, 3H), 1.27 (s, 3 H), 1.22 (s, 3H), 1.16 (s, 3H), 1.10 (s, 3H), 1.06 (s, 3H), 0.85 (s, 3H). The spectroscopic data agrees with previously reported data [57].

#### 3.2.2. General Procedure for Fischer Indolization (Compounds **4a**–**f**) 

A mixture of ketone **2** (0.42 mmol), phenylhydrazine (0.51 mmol) or *p*-substituted phenylhydrazine hydrochloride (0.51 mmol), and glacial acetic acid (4 mL) was heated at reflux during 2 h. When the mixture reached room temperature, the solid formed was collected by filtration, washed with cold glacial acetic acid and water. These compounds were pure enough and no additional purification steps were needed.

##### Compound **4a**

Yield 55%; m.p. 245–248 °C; ^1^H NMR (600 MHz, CDCl_3_): δ_H_ = 8.82 (bs, 1H), 7.36 (d, *J* = 7.7 Hz, 1H), 7.20 (d, *J* = 8.0 Hz, 1H), 6.92 (d, *J* = 7.9 Hz, 1H), 5.68 (s, 1H), 3.83 (d, *J* = 15.4 Hz, 1H), 2.57 (s, 1H), 2.19–2.13 (m, 2H), 2.10–1.47 (m, 16H), 1.33 (s, 3H), 1.24 (s, 3H), 1.16 (s, 3H), 1.12 (s, 3H), 1.10 (s, 3H), 1.07 (s, 3H), 0.75 (s, 3H). ^13^C NMR (151 MHz, CDCl_3_): δ_C_ = 199.9, 173.6, 169.7, 140.5, 136.2, 128.5, 127.9, 120.4, 118.2, 118.1, 110.3, 106.3, 60.3, 52.9, 48.4, 45.2, 43.4, 43.2, 41.1, 37.9, 37.7, 37.5, 33.9, 31.9, 31.7, 30.9, 30.8, 28.5, 28.4, 26.4, 26.3, 23.2, 23.1, 18.4, 18.2, 15.9. ESI–MS for C_36_H_48_NO_3_ [M+H]^+^: calc. 542.36 found 542.33; UPLC purity of >95.0% (retention time = 4.8 min, isocratic elution of 80% CH_3_CN/ 20% aqueous formic acid 0.1%, Acquity UPLC HSS T3 column).

##### Compound **4b**

Yield 53%; m.p. 243–244 °C; ^1^H NMR (600 MHz, CDCl_3_): δ_H_ = 7.66 (bs, 1H), 7.17 (d, *J* = 8.7 Hz, 1H), 6.97 (d, *J* = 2.3 Hz, 1H), 6.76 (dd, *J* = 8.7, 2.4 Hz, 1H), 5.80 (s, 1H), 3.92 (d, *J* = 15.4 Hz, 1H), 3.84 (s, 3H), 2.68 (s, 1H), 2.27–2.21 (m, 2H), 2.10–1.47 (m, 16H), 1.42 (s, 3H), 1.31 (s, 3H), 1.24 (s, 3H), 1.24 (s, 3H), 1.22 (s, 3H), 1.07 (s, 1H), 1.05 (d, *J* = 1.9 Hz, 1H), 0.87 (s, 3H). ^13^C NMR (151 MHz, CDCl_3_): δ_C_ = 200.3, 182.3, 169.5, 141.2, 131.3, 128.9, 128.7, 111.2, 110.3, 111.0, 107.2, 100.6, 60.7, 56.1, 53.1, 48.2, 45.6, 44.0, 43.5, 40.9, 38.2, 37.9, 37.8, 34.2, 32.2, 32.0, 31.2, 31.0, 28.7, 28.6, 26.7, 26.6, 23.7, 23.5, 18.6, 18.5, 16.2. ESI–MS for C_37_H_50_NO_4_ [M+H]^+^: calc. 572.37 found 572.38; UPLC purity of >95.0% (retention time = 4.6 min, isocratic elution of 80% CH_3_CN/ 20% aqueous formic acid 0.1%, Acquity UPLC HSS T3 column).

##### Compound **4c**

Yield 61%; m.p. 268–271 °C; ^1^H NMR (600 MHz, CDCl_3_): δ_H_ = 8.89 (bs, 1H), 7.09 (d, *J* = 0.5 Hz, 1H), 7.03 (d, *J* = 8.1 Hz, 1H), 6.73 (dd, *J* = 8.2, 1.5 Hz, 1H), 5.61 (s, 1H), 3.71 (d, *J* = 15.4 Hz, 1H), 3.26 (s, 1H, Ar–CH_3_), 2.27–2.21 (m, 2H), 2.10–1.47 (m, 16 H), 1.42 (s, 3H), 1.31 (s, 3H), 1.24 (s, 3H), 1.28 (s, 3H), 1.17 (s, 3H), 1.10 (s, 3H), 1.06 (s, 3H), 1.00 (s, 3H), 0.70 (s, 3H). ^13^C NMR (151 MHz, CDCl_3_): δ_C_ = 200.3, 182.3, 169.4, 140.7, 134.9, 129.4, 129.0, 128.6, 122.9, 118.9, 110.3, 107.3, 61.0, 53.5, 48.6, 45.9, 44.3, 43.8, 41.4, 38.5, 38.2, 34.5, 32.6, 32.4, 31.5, 31.4, 29.1, 28.9, 27.1, 23.9, 23.8, 21.9, 19.0, 18.8, 16.5. ESI–MS for C_37_H_50_NO_3_ [M+H]^+^: calc. 556.80 found 556.41; UPLC purity of >95.0% (retention time = 6.2 min, isocratic elution of 80% CH_3_CN/20% aqueous formic acid 0.1%, Acquity UPLC HSS T3 column).

##### Compound **4d**

Yield 73%; m.p. 245–248 °C; ^1^H NMR (600 MHz, CDCl_3_): δ_H_ = 8.63 (bs, 1H), 7.13 (dd, *J* = 8.7 and 4.3 Hz, 1H), 7.04 (dd, *J* = 9.7 and 2.5 Hz, 1H), 6.73 (ddd, *J* = 18.0, 9.0 and 2.5 Hz, 1H), 5.72 (s, 1H), 3.80 (d, *J* = 15.4 Hz, 1H), 2.59 (s, 1H), 2.08–1.34 (m, 16H) 1.28 (s, 1H), 2.27–2.21 (m, 2H), 1.42 (s, 3H), 1.31 (s, 3H), 1.24 (s, 3H), 1.28 (s, 3H), 1.17 (s, 3H), 1.10 (s, 3H), 1.07 (m, 1H), 1.06 (s, 3H), 1.00 (s, 3H), 0.70 (s, 3H). ^13^C NMR (151 MHz, CDCl_3_): δ_C_ = 200.3, 182.7, 169.6, 159.3, 156.9, 142.6, 133.1, 129.3, 111.1, 109.5, 109.3, 108.1, 104.2, 60.9, 53.4, 48.6, 45.9, 44.3, 43.8, 41.4, 38.5, 38.2, 38.0, 34.6, 32.5, 31.5, 29.0, 28.9, 27.0, 24.0, 23.8, 19.0, 18.8, 16.5. ESI–MS for C_36_H_47_FO_3_ [M+H]^+^: calc. 560.77 found 560.37; UPLC purity of >95.0% (retention time = 5.1 min, isocratic elution of 80% CH_3_CN/20% aqueous formic acid 0.1%, Acquity UPLC HSS T3 column).

##### Compound **4e**

Yield 70%; m.p. 256–258 °C; ^1^H NMR (600 MHz, CDCl_3_): δ_H_ = 9.38 (bs, 1H), 7.24 (d, 2.1 Hz, 1H), 7.06 (d, *J* = 8.5 Hz, 1H), 6.83 (dd, *J* = 8.5 and 2.1 Hz, 1H), 5.61 (s, 1H), 3.71 (d, *J* = 15.4 Hz, 1H), 2.50 (s, 1H), 1.28 (s, 1H), 2.27–2.21 (m, 2H), 2.07–1.45 (m, 17H), 1.28 (s, 3H), 1.19 (s, 3H), 1.11 (s, 3H), 1.06 (s, 3H), 1.04 (s, 3H), 1.00 (s, 3H), 0.70 (s, 3H). ^13^C NMR (151 MHz, CDCl_3_): δ_C_ = 199.8, 180.8, 169.2, 141.9, 134.7, 129.0, 124.8, 121.3, 118.3, 111.2, 107.4, 60.6, 53.6, 53.1, 48.3, 45.5, 43.9, 43.5, 41.2, 38.2, 37.9, 37.6, 34.3, 32.1, 31.2, 28.7, 28.6, 26.7, 26.6, 23.6, 23.5, 20.5, 18.7, 18.5, 16.1. ESI–MS for C_36_H_47_ClNO_3_ [M+H]^+^: calc. 576.32 found 576.31; UPLC purity of >95.0% (retention time = 6.5 min, isocratic elution of 80 % CH_3_CN/20% aqueous formic acid 0.1%, Acquity UPLC HSS T3 column). 

##### Compound **4f**

Yield 69%; m.p. 267–269 °C; ^1^H NMR (400 MHz, CDCl_3_): δ_H_ = 7.91 (bs, 1H, N*H*), 7.79 (s, 1H, H–4′), 7.34 (s, 2H, H–6′and H–7′), 5.80 (s, 1H, H–12), 3.97 (d, *J* = 15.7 Hz, 1H, 1α–H), 2.67 (s, 1H, H–9α), 2.29 (m, 1H, 1β–H), 2.25 (m, 1H, H–18β), 2.08 (m, 1H, H–15), 2.06 (m, 1H, H–16), 1.80 (m, 1H, H–7), 2.02 (m, 1H, H–19), 1.79 (m, 1H, H–21), 1.72 (m, 1H, H–6), 1.69 (m, 1H, H–19), 1.55 (m, 2H, H–7 and H–21), 1.44 (m, 2H, H–5α and H–22), 1.42 (s, 3H, 29–CH_3_), 1.39 (m, 1H, H–6), 1.33 (s, 3H, 24–CH_3_), 1.28 (m, 1H, H–16), 1.27 (s, 3H, 23–CH_3_), 1.24 (s, 3H, 26–CH_3_), 1.23 (s, 3H, 27–CH_3_), 1.18 (s, 3H, 25–CH_3_), 1.07 (m, 1H, H–16), 0.87 (s, 3H, 28–CH_3_). ^13^C NMR (151 MHz, CDCl_3_): δ_C_ = 199.8 (C11), 182.1 (C30), 169.3 (C13), 142.1 (C3), 137.8 (C3a’), 129.0 (C12), 127.9 (C7a’), 126.1 (CF_3_), 121.6 (C5′), 118.0 (C6′), 116.4 (C4′), 110.4 (C7′), 108.4 (C2), 60.5 (C9), 53.1 (C5), 48.3 (C18), 45.5 (C8), 44.0 (C14), 43.5 (C20), 41.1 (C19), 38.1 (C10), 37.9 (C22), 37.5 (C1), 34.2 (C4), 32.2 (C7), 31.1 (C17), 28.7 (C26), 28.6 (C28), 27.7 (C15), 26.6 (C16), 23.6 (C29), 23.5 (C23), 18.7 (C6), 18.5 (C27), 16.2 (C25). ESI–MS for C_37_H_47_F_3_NO_3_ [M+H]^+^: calc. 610.77 found 610.36; UPLC purity of >95.0% (retention time = 7.0 min, isocratic elution of 80% CH_3_CN/20% aqueous formic acid 0.1%, Acquity UPLC HSS T3 column). 

#### 3.2.3. Synthesis of 3,11-Dioxo-2-formylolean-12-en-30-oic Acid (**3**)

Under nitrogen atmosphere, compound **2** (500 mg, 1.1 mmol) was dissolved in dry dioxane (10 mL) and NaH (10 mmol) was slowly added. This mixture was heated to 45 °C; subsequently, ethyl formate (0.5 mL, 5.8 mmol) was added dropwise. After 45 min, the reaction mixture was refluxed for 5 h then cooled at room temperature and finally poured in 10% HCl (100 mL). The resulting precipitate was collected by vacuum filtration. The crude product was purified by flash chromatography using hexanes/ethyl acetate (80:20) to give 260 mg (50%) of **3** as colorless crystals. M.p: 260–263 °C. ^1^H NMR (600 MHz, CDCl_3_): δ_H_ = 8.64 (d, *J* = 2.9 Hz, 1H), 5.78 (s, 1H), 3.46 (d, *J* = 14.8 Hz, 1H), 2.44 (s, 1H), 2.23 (dd, *J* = 13.4, 3.5 Hz, 1H), 1.94 (d, *J* = 15 Hz, 1H), 1.86 (td *J* = 13.6, 4.4 Hz, 1H), 1.63 (t, *J* = 13.6 Hz, 1H), 1.38 (s, 3H), 1.23 (s, 3H), 1.20 (s, 3H), 1.17 (s, 3H), 1.14 (s, 3H), 1.13 (s, 3H), 0.86 (s, 3H). The spectroscopic data agrees with previously reported data [58].

#### 3.2.4. Synthesis of *N*-Phenylpyrazole Derivatives **5a**–**f**

To a solution of compound **3** (150 mg, 0.3 mmol) in dry ethanol (5 mL), phenylhydrazine (0.45 mmol) or *p*-substituted phenylhydrazines hydrochloride (0.45 mmol) was added. This solution was stirred at 70 °C for 1 h, cooled at room temperature and the solvent was subsequently evaporated. The crude product was purified by flash chromatography using hexanes/ethyl acetate (from 80:20 to 70:30). Afterwards, the resulting compounds were recrystallized from CH_2_Cl_2_ and MeOH. 

##### Compound **5a**

Yield 32%; light yellow solid; m.p. 243–245 °C; ^1^H NMR (600 MHz, CDCl_3_): δ_H_ = 7.43 (m, 3H), 7.39 (m, 3H), 5.77 (s, 1H), 3.76 (d, *J* = 15.4 Hz, 1H), 2.55 (s, 1H), 2.04–1.21 (m, 17H), 1.38 (s, 3H), 1.19 (s, 3H), 1.18 (s, 3H), 1.16 (s, 3H), 1.06 (s, 3H), 1.03 (s, 3H), 0.85 (s, 3H). ^13^C NMR (151 MHz, CDCl_3_): δ_C_ = 200.0, 181.3, 169.7, 145.8, 142.3, 138.4, 129.3, 128.8, 114.4, 60.6, 60.5, 54.5, 48.3, 45.3, 43.9, 43.4, 41.1, 38.1, 37.8, 37.7, 34.9, 32.1, 32.0, 31.0, 29.5, 28.7, 28.6, 26.7, 26.5, 23.4, 22.6, 21.2, 18.5, 18.3, 15.7, 14.3. ESI–MS for C_37_H_48_N_2_O_3_ [M+H]^+^: calc. 569.80 found 569.32; UPLC purity of >95.0% (retention time = 6.3 min, isocratic elution of 80% CH_3_CN/20% aqueous formic acid 0.1%, Acquity UPLC HSS T3 column).

##### Compound **5b**

Yield 44%; colorless crystals; m.p. 208–211 °C; ^1^H NMR (400 MHz, CDCl_3_): δ_H_ = 7.37 (s, 1H), 7.30 (d, *J* = 8.5 Hz, 2H), 6.92 (d, *J* = 8.6 Hz, 2H), 5.77 (s, 1H), 3.85 (s, 3H), 3.76 (d, *J* = 15.4 Hz, 1H), 2.55 (s, 1H), 2.26–1.41 (m, 17H), 1.39 (s, 3H), 1.22 (s, 3H), 1.19 (s, 3H), 1.16 (s, 3H), 1.07 (s, 3H), 1.03 (s, 3H), 0.86 (s, 3H). ^13^C NMR (151 MHz, CDCl_3_): δ_C_ = 199.9, 181.1, 169.7, 159.9, 145.9, 138.3, 135.2, 130.4, 128.9, 114.3, 113.7, 60.6, 55.7, 54.6, 48.3, 47.0, 45.0, 45.4, 43.9, 43.5, 42.2, 41.2, 38.2, 37.9, 37.8, 34.9, 32.2, 32.1, 31.1, 29.5, 28.8, 28.6, 26.7, 26.6, 23.4, 22.6, 18.4, 15.7. ESI–MS for C_38_H_81_N_2_O_4_ [M+H]^+^: calc. 599.80 found 599.39; UPLC purity of >95.0% (retention time = 5.7 min, isocratic elution of 80% CH_3_CN/20% aqueous formic acid 0.1%, Acquity UPLC HSS T3 column).

##### Compound **5c**

Yield 38%; colorless crystals; m.p. 240–241 °C; ^1^H NMR (400 MHz, CDCl_3_): δ_H_ = 7.36 (s, 1H), 7.28 (d, *J* = 8.2 Hz, 2H), 7.21 (d, *J* = 8.2 Hz, 2H), 5.78 (s, 1H), 3.76 (d, *J* = 15.4 Hz, 1H), 2.55 (s, 1H), 2.41 (s, 3H), 2.26–1.40 (m, 17H), 1.39 (s, 3H), 1.22 (s, 3H), 1.19 (s, 3H), 1.16 (s, 3H), 1.07 (s, 3H), 1.03 (s, 3H), 0.86 (s, 3H). ^13^C NMR (151 MHz, CDCl_3_): δ_C_ = 200.2, 181.4, 170.0, 146.1, 140.2, 139.3, 138.7, 129.5, 129.4, 129.2, 114.7, 60.9, 54.9, 48.7, 45.7, 44.2, 43.7, 41.5, 38.5, 38.2, 38.1, 35.2, 32.5, 32.4, 31.4, 29.9, 29.1, 28.9, 27.1, 26.9, 23.7, 22.9, 21.7, 18.9, 18.7, 16.1. ESI–MS for C_38_H_51_N_2_O_3_ [M+H]^+^: calc. 583.83 found 583.39; UPLC purity of >95.0% (retention time = 8.7 min, isocratic elution of 80% CH_3_CN/20% aqueous formic acid 0.1%, Acquity UPLC HSS T3 column). 

##### Compound **5d**

Yield 42%; white solid; m.p. 260–264 °C; ^1^H NMR (400 MHz, CDCl_3_): δ_H_ = 7.37 (m, 3H), 7.11 (t, *J* = 8.4 Hz, 2H), 5.78 (s, 1H), 3.77 (d, *J* = 15.4 Hz, 1H), 2.56 (s, 1H), 2.09–1.23 (m, 17H), 1.39 (s, 3H), 1.22 (s, 3H), 1.19 (s, 3H), 1.16 (s, 3H), 1.07 (s, 3H), 1.03 (s, 3H), 0.86 (s, 3H). ^13^C NMR (151 MHz, CDCl_3_): δ_C_ = 199.8, 181.1, 180.1, 164.0, 161.5, 146.0, 138.8, 131.2, 131.1, 128.9, 115.7, 115.4, 114.7, 60.6, 54.6, 48.4, 45.4, 43.9, 43.5, 41.2, 38.2, 37.9, 37.8, 34.9, 32.2, 32.1, 31.1, 29.6, 28.8, 28.6, 26.7, 26.6, 23.4, 22.7, 18.6, 18.4, 15.8. ESI–MS for C_37_H_48_FN_2_O_3_ [M+H]^+^: calc. 587.79 found 587.36; UPLC purity of >95.0% (retention time = 6.2 min, isocratic elution of 80% CH_3_CN/20% aqueous formic acid 0.1%, Acquity UPLC HSS T3 column).

##### Compound **5e**

Yield 70%; colorless crystals; m.p. 263–266 °C; ^1^H NMR (400 MHz, CDCl_3_): δ_H_ = 7.41 (d, *J* = 8.4 Hz, 2H), 7.37 (s, 1H), 7.33 (d, *J* = 8.5 Hz, 2H), 5.78 (s, 1H), 3.77 (d, *J* = 15.4 Hz, 1H), 2.56 (s, 1H), 2.13–1.23 (m, 17H), 1.39 (s, 3H), 1.22 (s, 3H), 1.19 (s, 3H), 1.16 (s, 3H), 1.07 (s, 3H), 1.03 (s, 3H), 0.86 (s, 3H). ^13^C NMR (151 MHz, CDCl_3_): δ_C_ = 200.2, 181.5, 181.4, 170.0, 146.4, 141.3, 139.3, 135.3, 131.0, 129.2, 115.1, 60.9, 54.9, 48.7, 45.7, 44.3, 43.8, 41.5, 38.5, 38.2, 38.0, 35.2, 32.5, 32.4, 31.5, 31.4, 29.9, 29.1, 28.9, 27.1, 26.9, 23.7, 23.1, 18.9, 18.7, 16.1. ESI–MS for C_37_H_48_ClN_2_O_3_ [M+H]^+^: calc. 603.24 found 603.34; UPLC purity of >95.0% (retention time = 9.5 min, isocratic elution of 80% CH_3_CN/20% aqueous formic acid 0.1%, Acquity UPLC HSS T3 column).

##### Compound **5f**

Yield 84%; m.p. 270–271 °C; ^1^H NMR (400 MHz, DMSO-*d*_6_): δ_H_ = 7.86 (d, *J* = 8.1 Hz, 2H, H–2′ and H–6′), 7.61 (d, *J* = 8.0 Hz, 2H, H–3′ and H–5′), 7.36 (s, 1H, pyrazole–H), 5.47 (s, 1H, H–12), 3.55 (d, *J* = 15.6 Hz, 1H, 1β–H), 2.57 (s, 1H, H–9), 2.17 (d, *J* = 15.4 Hz, 1H, H–1α), 2.11 (m, 1H, H–19), 2.10 (m, 1H, H–16), 2.06 (m, 1H, H–18), 1.83 (m, 1H, H–21), 1.80 (m, 1H, H–15), 1.70 (m, 1H, H–7), 1.69 (m, 1H, H–19), 1.68 (m, 1H, H–6), 1.52 (m, 1H, H–6), 1.42 (m, 1H, H–7), 1.36 (m, 1H, H–22), 1.35 (m, 1H, H–15), 1.35 (s, 3H, 29–CH_3_), 1.33 (m, 1H, H–21), 1.32 (m, 1H, H–5), 1.25 (m, 1H, H–22), 1.08 (s, 3H, 26–CH_3_), 1.08 (s, 3H, 27–CH_3_), 0.99 (m, 1H, H–16), 0.97 (s, 3H, 23–CH_3_), 0.94 (s, 3H, 24–CH_3_), 0.75 (s, 3H, 28–CH_3_). ^13^C NMR (151 MHz, CDCl_3_): δ_C_ = 198.8 (C11), 177.8 (C30), 170.4 (C13), 145.9(C3a’), 145.8 (C3), 138.7 (C–pyrazole), 130.2 (C5′), 129.74 (C7′), 127.5 (C12), 126.2 (C6′), 126.1 (C4′), 126.08 (C7a’), 122.75 (CF_3_), 114.40 (C2) 59.8 (C9), 53.6 (C5), 48.25 (C18), 44.8 (C14), 44.7 (C8), 43.3 (C20), 41.02 (C19) 37.8 (C10), 37.7 (C22), 37.03 (C1), 34.5 (C4), 31.8 (C7), 31.5 (C17), 30.6 (C21), 29.5 (C23), 28.2 (C28), 28.0 (C27), 26.40 (C15), 26.0 (C16), 23.1(C29), 22.7 (C24), 18.1 (C26), 18.05 (C6), 15.63 (C25). ESI–MS for C_38_H_48_ F_3_N_2_O_3_ [M+H]^+^: calc. 637.80 found 637.41; UPLC purity of >95.0% (retention time = 8.7 min, isocratic elution of 80% CH_3_CN/20% aqueous formic acid 0.1%, Acquity UPLC HSS T3 column).

### 3.3. In Vitro Assays

#### 3.3.1. PTP1B Inhibitory Assay

To test the inhibitory activity of the compounds against PTP1B, a spectrocolorimetric method previously described was employed [59,60]. The compounds and positive control were dissolved in DMSO or buffer solution [50 mM; Tris-HCI Buffer, pH 6.8], DL-dithiothreitol 1.5 mM, all from Sigma–Aldrich, St. Louis, MO, USA. Aliquots of 0–10 μL of testing compounds (triplicated) were incubated for 10 min at 37 °C with 85 μL of enzyme stock solution (3 μM) in buffer solution. After incubation, 5 μL of the substrate *p*-nitrophenyl phosphate (pNPP, 0.5 mM, Sigma–Aldrich, St. Louis, MO, USA) were added and incubated a further 25 min at 37 °C; then, the absorbance was recorded at 415 nm.

The IC_50_ was calculated by regression analysis using Equation:$$\%PTP1B=\frac{A_{100}}{1+{\left(\frac{i}{{\mathrm{IC}}_{50}}\right)}^{s}}$$
where %*PTP*1*B* is the percentage of inhibition, A_100_ is the maximum inhibition, *i* is the inhibitor concentration, IC_50_ is the concentration required to inhibit the activity of the enzyme by 50%, and *s* is the cooperative degree.

#### 3.3.2. Enzyme Kinetics

The kinetic studies were carried out under the same conditions described above. Different saturation curves were constructed at fixed inhibitor concentrations. The different inhibitor concentrations were determined based on the previously determined IC_50_ for each compound. Subsequently, the data obtained were analyzed through a non-linear regression, using OriginPro version 2018 (64-bit) SR1 (Northamton MA, USA, https://www.originlab.com) (accessed on 2 April 2021), adjusting to the following mathematical models: 

Competitive:$$y=V_{m}\times \frac{s}{K_{m}\times \left(1+\frac{i}{K_{ic}}\right)+s\times \left(1+\frac{i}{K_{iu}}\right)}$$
where *V_m_* is the maximum velocity, *S* is the substrate, *K_m_* is the Michaelis–Menten constant, *K_ic_* and *K_iu_* are the competitive and uncompetitive inhibition constants, respectively.

And non-competitive:$$y=V_{m}\times \frac{s^{nh}}{\left(1+\frac{i}{K_{i}}\right)\times S_{05}{}^{nh}+\left(1+\frac{i}{K_{i}}\right)\times s^{nh}}$$
where *V_m_* is the maximum velocity, *S* and *i* are the concentration of substrate and inhibitor, respectively; *K_i_* is the inhibition constant, and *nh* is the number of Hill.

The type of inhibition of PTP1B was determined by fitting the data to both models, where the fit with the best *R*^2^ indicates the type of inhibition of the compounds. Additionally, Lineweaver–Burk graphs were constructed to visualize the type of inhibition graphically [61].

### 3.4. In Silico Studies

Ligands were constructed and their geometry was optimized employing the semi-empirical Parameterization Method 3 (PM3) using Chem3D BioUltra 16.0 software (PerkinElmer, Waltham, MA, USA) with GAMESS interface [62]. X–ray structures of PTP1B were retrieved from the Protein Data Bank (http://www.rcsb.org/) (accessed on 10 April 2021) entries 1C83 and 1T49. Using YASARA structure (version 20.7.43) [54], first the water molecules were removed from the macromolecules, then their geometry was minimized employing the NOVA [63] forcefield by running the *em_clean* macro. After removing the co-crystallized ligand, the minimized structures (saved as *.pdb format) were used to dock each ligand employing Autodock 4.2 (The Scripps Research Institute, La Jolla, CA, USA) [64], Autodock Vina (The Scripps Research Institute, La Jolla, CA, USA) [65] and GOLD (The Cambridge Crystallographic Data Centre, Cambridge, UK) version 2020.2 [66]. On the other hand, MD simulations of ligand–protein complexes were performed using YASARA structure. The results from docking and MD simulations were visualized using PyMOL (The PyMOL Molecular Graphics System, Version 2.0 Schrödinger, LLC) and Protein Ligand Interaction Profiler server [67,68] whereas 2D–interaction diagrams were produced with Discovery Studio visualizer 2021 (Dassault Systèmes, San Diego, CA, USA) [69]. The protocol for docking and MD simulations studies was as follows.

#### 3.4.1. Molecular Docking

For molecular docking using Autodock 4.2 and Autodock Vina, the graphical interface AutoDockTools 1.7.1 [64] suite was used to prepare and analyze the docking simulations. Hydrogen atoms were added to the macromolecules and Gasteiger–Marsilli charges were assigned to the atoms in the protein as well as ligands. Both protein and ligands were exported as *.pdbqt files. Docking simulations using Autodock 4.2 at the catalytic site of PTP1B (using 1C83 macromolecule) were performed with a grid box size: 45 Å × 60 Å × 47 Å with a spacing of 0.345 Å and coordinate: x = −21.39, y = 6.76 and z = −0.48; whereas docking simulations at the allosteric site of PTP1B (using 1T49 macromolecule) were performed with a grid box size: 60 Å × 60 Å × 60 Å with a spacing of 0.345 Å and coordinates x = 55.74, y = 33.41 and z = 24.47. The search was carried out with the Lamarckian Genetic Algorithm. A total of 100 GA runs with a maximal number of 25,000,000 evaluations, a mutation rate of 0.02 and an initial population of 150 conformers were covered. Finally, each ligand with the best cluster size and the lowest binding energy was selected for further analysis. Molecular docking using Autodock Vina was carried out employing the same coordinates as previously mentioned, except that the grid box dimensions were: 16.8 Å × 22.5 Å × 17.6 Å and 22.5 Å × 22.5 Å × 22.5 Å for 1C83 and 1T49 macromolecules, respectively, and the exhaustiveness value was set to 500.

The resulting ligand complexes obtained for both 1C83 and 1T49 macromolecules were exported to GOLD. Using the GOLD wizard, the proteins were prepared by adding hydrogens and extracting the ligands which were further docked at the catalytic site or allosteric site within a 6 Å radius sphere was carried out using the following parameters: 100 genetic algorithm runs and 125,000 operations. CHEMPLP fitness was chosen as the main scoring function whereas GoldScore fitness was chosen as the re-scoring function. The dockings were ranked according to the value of the CHEMPLP and GoldScore fitness function.

Using the LigRMSD web server (https://ligrmsd.appsbio.utalca.cl/) (accessed on 12 April 2021), the docking protocol for each software was validated by comparing the RMSD value of the co-crystal structure and the docked structure [70]. 

#### 3.4.2. MD Simulations 

The ligand–protein complexes were submitted to MD simulations with YASARA Structure [54]. The simulations started by running the *md_run* macro, which included an optimization of the hydrogen bonding network to increase the solute stability, and a pKa prediction to fine-tune the protonation states of protein residues at the chosen pH of 7.4. NaCl ions were added with a physiological concentration of 0.9%, with an excess of either Na or Cl to neutralize the cell. After steepest descent and simulated annealing minimizations to remove clashes, the simulation was run for 50 nanoseconds using the AMBER14 force field [55] for the solute, GAFF2 [71] and AM1BCC [72] for ligands and TIP3P for water. The cutoff was 8 Å for Van der Waals forces (the default used by AMBER [73], no cutoff was applied to electrostatic forces (using the Particle Mesh Ewald algorithm, [74]). The equations of motions were integrated with a multiple timestep of 1.25 fs for bonded interactions and 2.5 fs for non-bonded interactions at a temperature of 298 K and a pressure of 1 atm (NPT ensemble) using algorithms described in detail previously [54]. After inspection of the solute RMSD as a function of simulation time, the first 100 picoseconds were considered equilibration time and excluded from further analysis.

The binding energy and SMD studies were performed by running the *md_analyzebindingenergy* and *md_steered* macros, respectively. 

## 4. Conclusions

Two A–ring–fused GA series having an indole or *N*-phenylpyrazole moieties were prepared. These moieties dramatically improved from 6- to 25-fold the PTP1B inhibitory activity of GA. Interestingly, several of the titled compounds exhibited better potency than those of claramine and ursolic acid. However, adding a chlorine atom at the phenyl moiety in both series was found to have a negative impact on the inhibitory activity against PTP1B. Nevertheless, the presence of a trifluoromethyl group at the same position greatly improves this activity. The results regarding the PTP1B inhibitory effect along with the enzyme kinetics studies demonstrated that both **4f** and **5f** are the most potent non-competitive PTP1B inhibitors with IC_50_ values = 2.5 and 4.4 µM as well as Ki values = 3.9 and 4.6 µM, respectively. It is worthwhile to mention that 5f is the only compound that has a better inhibitory effect than that of suramin.

The docking simulations performed on PTP1B showed that the newly-synthesized compounds bind preferably to the allosteric binding site with lower binding free energies as well as better scores than that of ursolic acid (non-competitive inhibitor). In addition, these studies displayed that the π-stacking and fluorine (from –CF_3_) interactions performed by the substituted indole or *N*-phenylpyrazole moieties with the amino acid residues Phe^196^, Phe^280^, Glu^276^ and Ala^189^ located inside this enzyme cavity have a positive impact on the binding affinity. The results from MD simulations demonstrated that the binding of **4f** at the allosteric binding site stabilizes better the PTP1B in its inactive conformation compared to **5f** and ursolic acid. However, both **4f** and **5f** have a similar binding energy profile. Finally, the SMD study showed that ursolic acid has the slowest dissociation rate from this site, followed by **4f** and **5f**.

We look forward to expanding the library of these compounds with the purpose of elucidating quantitative structural relationships associated with their PTP1B inhibitory effect. Moreover, further research can be conducted to determine the anti-obesity and anti-diabetic effect in vivo for the titled compounds.

## Data Availability

The data presented in this study are contained within the article.

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
