# Peer review of "Indole- and Pyrazole-Glycyrrhetinic Acid Derivatives as PTP1B Inhibitors: Synthesis, In Vitro and In Silico Studies"

_molecules, 2021, doi:10.3390/molecules26144375_

Round 1

Reviewer 1 Report

This work is devoted to the synthesis and study of the effect of fragments of indole or N-phenylpyrazole in glycyrrhetinic acid on inhibition Protein tyrosine phosphatase 1B (PTPB1). Inhibition of this enzyme allows you to fight diseases such as diabetes, obesity, and Alzheimer's and Parkinson's diseases. The leading compounds in the ability to inhibit PTPB1B were identified. The docking performed on the same PTPB1 preferably binds to the allosteric binding site, and can also be conducted to determine the anti-obesity and antidiabetic effect in vivo for the titled compounds.

                The work is written in quite detail and in the smallest details. However, I would like to note some shortcomings in the work that do not allow it to be a fairly significant study. The conclusion of the work is quite compressed, increase the text of the comment on PTP1B inhibitory activity of indole-and pyrazole-GA derivatives and modeling of docking.

                As you can comment on several signals of impurities in the spectrum in upfield.(figure below) What is the value of their presence in the 1H NMR spectra? Try to explain why they are needed, otherwise they should not be.

Author Response

Sincerest thanks for your timely response comments on our manuscript. We sincerely apologize for the great time it has taken us to respond to these comments. Nevertheless, we have done our very best to enrich, clarify and complete the work herein and hope that the revised version of the manuscript can still be considered for publication in Molecules. We have modified the paper in response to the extensive reviewers’ comments: we have added 13C NMR data for the final compounds in the main text. Furthermore, some sections of the manuscript have been rewritten in hope that these comply with the referees’ remarks. We will respond to your comments pointcounterpoint.  

Comment 1:

The work is written in quite detail and in the smallest details. However, I would like to note some shortcomings in the work that do not allow it to be a fairly significant study. The conclusion of the work is quite compressed, increase the text of the comment on PTP1B inhibitory activity of indole-and pyrazole-GA derivatives and modeling of docking.

Response:

We agree with this comment, thus, the conclusion has been extended.

Comment 2:

As you can comment on several signals of impurities in the spectrum in upfield.(figure below) What is the value of their presence in the 1H NMR spectra? Try to explain why they are needed, otherwise they should not be.

Response

In response to this comment, the highlighted signals are not impurities, they correspond to some CH2 and CH signals of the triterpenoid skeleton. These signals have been reported in the main text. The signal at 0.0 ppm corresponds to TMS. Nevertheless, we noticed that the spectrum of compounds 4af showed some signals that correspond to acetic acid (the solvent that was used to obtain these compounds) while compounds 5e and 5f showed some signals that corresponds to methanol (the solvent used to recrystallize these compounds). A note was added to these signals in the spectra found in the Supplementary Materials.

Reviewer 2 Report

Ledy De-la-Cruz-Martínez and coworkers describe the synthesis and evaluation of two series of indole- and N-phenylpyrazole-GA (glycyrrethinic acid) derivatives as PTP1B inhibitors. The authors measured their inhibitory activity and enzyme kinetics against PTP1B and the trifluoromethyl derivative of indole-GA (4f) exhibited non-competitive inhibition of PTP1B as well as higher potency than that of positive controls. Thus, this study indicates that semisynthetic derivatives of GA may be developed as PTP1B inhibitors and can serve as promising lead compounds for any further pursuit in this area of research. Overall, the manuscript is well written and the rational of the study is clearly presented. Based on the novelty and originality of the manuscript, I think this manuscript should be accepted with positive revision for publication in Molecules.

The manuscript could be publishable after addressing the following questions:

  1. First, the authors have not demonstrated that why they combine the scaffolds of indole and GA. Then, according to the reference 43 given in the article, a para- pyridine substituted maslinic acid was the most active compound towards PTP1B with an IC50 value of 64μM. Besides, 3,4-dimethyl and 2- chloro- phenyl substituted lithocholic acid derivatives exhibited similar PTP1B inhibitory activities with IC50 values of 0.73μM and 0.86μM, respectively (the reference 45). The authors have not demonstrated that why they did not consider such substituents. Thus, the authors' design have not sufficient scientific basis. We hope that the design part can be fully justified.
  2. There are some errors in typos, consistency in the manuscript. Page 4 line 123, Page 5 line 146/ 148 and Page 11 line 317, 5g should be 5f. Besides, Page 10 line 277, 4g should be 4f. In Fig S21, “1H NMR (400 MHz) spectrum of compound 5e in DMSO-d6”. However, we found that this is a 1H NMR (400 MHz) spectrum of compound 5f. Therefore the authors need to correct these errors.
  3. In this manuscript, the authors have summarized that “ Interestingly, this triterpene can cross the blood brain barrier which is an essential property in the context of PTP1B inhibitors because reducing the hypothalamic PTP1B activity of obese mice has shown to greater enhance the leptin and insulin sensitivity than that of other tissues, which also effectively decreases adiposity and improves glucose metabolism”. Have the authors analyzed the difference of the permeability of the synthesized compounds to the blood-brain barrier in vivo and the elimination of the tissue content with the increase of time?

Author Response

Sincerest thanks for your timely response comments on our manuscript. We sincerely apologize for the great time it has taken us to respond to these comments. Nevertheless, we have done our very best to enrich, clarify and complete the work herein and hope that the revised version of the manuscript can still be considered for publication in Molecules. We have modified the paper in response to the extensive reviewers’ comments. we have added 13C NMR data for the final compounds in the main text. Furthermore, some sections of the manuscript have been rewritten in hope that these comply with the referees’ remarks. We will respond to your comments point–counterpoint. 

Comment 1:

First, the authors have not demonstrated that why they combine the scaffolds of indole and GA. Then, according to the reference 43 given in the article, a para- pyridine substituted maslinic acid was the most active compound towards PTP1B with an IC50 value of 64μM. Besides, 3,4-dimethyl and 2- chloro- phenyl substituted lithocholic acid derivatives exhibited similar PTP1B inhibitory activities with IC50 values of 0.73μM and 0.86μM, respectively (the reference 45). The authors have not demonstrated that why they did not consider such substituents. Thus, the authors' design have not sufficient scientific basis. We hope that the design part can be fully justified.

Response

In response to this comment, we deleted references 44 and 45 since they refer to lithocholic acid which is not an oleanane-type triterpene. We based our design in the maslinic acid (a pentacyclic oleanane-type triterpene) derivatives (reference 43), which are structurally more related to the compounds presented in our study. We agree with your comment regarding introducing substituents in another positions. However, when we have employed meta-phenyhydrazines, we obtained isomeric mixtures of GA-indole derivatives which are very difficult to purify.

In section 1, we have included a paragraph describing the basis of our design.   

Comment 2:

There are some errors in typos, consistency in the manuscript. Page 4 line 123, Page 5 line 146/ 148 and Page 11 line 317, 5g should be 5f. Besides, Page 10 line 277, 4g should be 4f. In Fig S21, “1H NMR (400 MHz) spectrum of compound 5e in DMSO-d6”. However, we found that this is a 1H NMR (400 MHz) spectrum of compound 5f. Therefore the authors need to correct these errors.

Response

We agree with this comment, the typos have been corrected. We have carefully read the text and have improved the English grammar.

Comment 3:

In this manuscript, the authors have summarized that “ Interestingly, this triterpene can cross the blood brain barrier which is an essential property in the context of PTP1B inhibitors because reducing the hypothalamic PTP1B activity of obese mice has shown to greater enhance the leptin and insulin sensitivity than that of other tissues, which also effectively decreases adiposity and improves glucose metabolism”. Have the authors analyzed the difference of the permeability of the synthesized compounds to the blood-brain barrier in vivo and the elimination of the tissue content with the increase of time?

Response

The cited paragraph refers to studies previously reported for GA. The paragraph has been rephrased. Since we do not have the infrastructure to carry out in vivo studies, we have not performed the aforementioned experiments. However, we appreciate your input, and we hope to be able to include these studies in the future.

Reviewer 3 Report

This work shows the synthesis of heterocyclic derivatives of glycyrrethinic acid and their role as PTP1B inhibitors, with promising results about the inhibition of this protein which is relevant in diabetes and obesity. The methodology and results are sound, with proper conclusions. To improve the quality of the manuscript, I consider that the following issues must be addressed:

  • An additional review of English language semantics and grammar is recommended, as there are some writing errors in some paragraphs.
  • Be careful in the typographical errors of some of the units of the quantities used (e. g. mg/kg)
  • Was a blind docking performed to find the allosteric site to explore another binding site different to that is found in PDB 1T49? It would be interesting due to the slight differences between the docking scores of the allosteric site and the catalytic site explored in this work.
  • The MD and SMD simulations were performed with 5g or 5f, or both? In Figure 5, the molecule denoted is 5f, whereas in Figure 6 is 5g.
  • The authors claim that UA showed “poorer” binding energy than compounds 4f and 5f according to the steered-molecular dynamics despite of the average energy goes in contradiction with this statement. A more in-depth discussion on how the SMD simulations were performed would be worthwhile; since, based on the magnitude and sign of the binding energy, it would suggest that UA has a higher affinity than the other compounds. How would this binding energy be related to what was observed in the PTP1B inhibition tests?
  • According to Figure 6B, could UA be considered a slow-binding molecule and 5f as a fast-binding molecule? Could this data be related with a possible residence time of the molecules and the inhibition observed in PTP1B?

Author Response

Sincerest thanks for your timely response comments on our manuscript. We sincerely apologize for the great time it has taken us to respond to these comments. Nevertheless, we have done our very best to enrich, clarify and complete the work herein and hope that the revised version of the manuscript can still be considered for publication in Molecules. We have modified the paper in response to the extensive reviewers’ comments: we have added 13C NMR data for the final compounds in the main text. Furthermore, some sections of the manuscript have been rewritten in hope that these comply with the referees’ remarks. We will respond to your comments point–counterpoint.  

Comment 1:

An additional review of English language semantics and grammar is recommended, as there are some writing errors in some paragraphs. Be careful in the typographical errors of some of the units of the quantities used (e. g. mg/kg)

Response

We have carefully read the text and have improved the English grammar. The writing errors have been corrected.

Comment 2:

Was a blind docking performed to find the allosteric site to explore another binding site different to that is found in PDB 1T49? It would be interesting due to the slight differences between the docking scores of the allosteric site and the catalytic site explored in this work.

Response

Your point is well made, however, we have done previously a blind docking on the 1T49 macromolecule and we noticed that our compounds bind to other sites weaker than the allosteric site.

In addition, we carried out docking simulations in the allosteric cavity using the 1T49 structure because this structure has a experimentally validated allosteric site. Thus, it has demostrated that the binding of molecules at this cavity inhibits the PTP1B's activity.

Comment 3:

The MD and SMD simulations were performed with 5g or 5f, or both? In Figure 5, the molecule denoted is 5f, whereas in Figure 6 is 5g.

Response

In response to this comment, the MD and SMD simulations were performed with 5f. The text in figures 6A and 6B has been corrected.

Comment 4:

The authors claim that UA showed “poorer” binding energy than compounds 4f and 5f according to the steered-molecular dynamics despite of the average energy goes in contradiction with this statement. A more in-depth discussion on how the SMD simulations were performed would be worthwhile; since, based on the magnitude and sign of the binding energy, it would suggest that UA has a higher affinity than the other compounds. How would this binding energy be related to what was observed in the PTP1B inhibition tests?

Response

In response to this comment, we have discussed indepth the MD and SMD simulations in sections 2.5.1 and 2.5.2. The results for the binding energies provided by YASARA software indicate that more positive values, correspond to better binding in the context of the forcefield used. Please find a more detailed explanations in the following link: http://shaker.umh.es/yas/20.10.4/BindEnergyObj.html    

Comment 5:

According to Figure 6B, could UA be considered a slow-binding molecule and 5f as a fast-binding molecule? Could this data be related with a possible residence time of the molecules and the inhibition observed in PTP1B?

Response

We have added in section 2.5.2 a more detailed explanation where we discuss the residence time of compounds and the inhibitory effect against PTP1B.

Reviewer 4 Report

This manuscript is a well-executed study investigating indole- and pyrazole-glycyrrhetinic acid derivatives, their biological activities. I find the manuscript quite rational, and recommend publication with following minor changes:

  1. Please check the spelling of "glycyrrhetinic" in the title.
  2. I do not see the need for Table 1 in the text.  It can be either moved to Supporting Material, or totally removed, since the NMR spectra for all compounds are already described under experimental protocols. Description of NMR peak assignments and explanations are quite good as they are, even without the Table 1.

Author Response

Sincerest thanks for your timely response comments on our manuscript. We sincerely apologize for the great time it has taken us to respond to these comments. Nevertheless, we have done our very best to enrich, clarify and complete the work herein and hope that the revised version of the manuscript can still be considered for publication in Molecules. We have modified the paper in response to the extensive reviewers’ comments. We have added 13C NMR data for the final compounds in the main text. Furthermore, some sections of the manuscript have been rewritten in hope that these comply with the referees’ remarks. We will respond to your comments point–counterpoint.  

Comment 1:

Please check the spelling of "glycyrrhetinic" in the title.

Response

We have carefully read the text and have improved the English grammar. The writing errors have been corrected including the spelling of "glycyrrhetinic" in the title.

Comment 2:

I do not see the need for Table 1 in the text.  It can be either moved to Supporting Material, or totally removed, since the NMR spectra for all compounds are already described under experimental protocols. Description of NMR peak assignments and explanations are quite good as they are, even without the Table 1.

Response

We agree with this comment, Table 1 has been moved to Supplementary Materials section.

Round 2

Reviewer 3 Report

This reviewed paper shows major improvements from the last submission. All the comments previously made have been addressed; thus, I recommend this paper for publication.

Although, there is a brief comment about the binding energy calculated by SMD simulations. Yasara is not very clear about the energies' meaning, especially the sign, and this tends to confuse the readers, but once the binding equilibria equation is planted, the sign has more meaning. For future works, I recommend that if this computational experiment is going to be reported, is better to interpret the results based on the equilibria equation of the experiment.